# PreCoT: Problem Representation Enhances Reasoning in Large Language Models

## Abstract

Chain-of-Thought (CoT) prompting has broken new ground in exploring the reasoning capabilities of large language models (LLMs). Recent studies expand this direction to specific strategies, such as question decomposition and planning, to improve the solution process. On the other hand, within cognitive psychology, problem representation construction is considered a vital aspect of human problem-solving along with the solution process. It involves a solver structurally encoding a problem by defining its initial and goal states, thereby improving the solution process. However, the construction of problem representation has yet to be tapped in further exploring the potential of LLMs' human-like problem-solving ability. In this work, we propose **P**roblem **R**epresentation **E**nhanced **CoT** (PreCoT), a novel prompting framework that enhances the solution process of LLMs with problem representation. PreCoT is divided into two main stages. First, it extracts the ingredients of the initial and goal state of the problem, which constitute the problem representation together. Next, it initiates an enhanced solution process based on the generated problem representation. In extensive evaluation on benchmarks from a wide range of domains, including arithmetic, commonsense, and symbolic reasoning, PreCoT outperforms CoT on most tasks in both few-shot and zero-shot manners. Additional analyses further demonstrate the effectiveness of problem representation and its contribution to the reasoning in LLMs, such as robustness against irrelevant context and problem context sensitivity.

## 1 Introduction

Scaling up language model (LM) has been a pivotal strategy to yield exceptional results across numerous Natural Language Processing (NLP) benchmarks (Devlin et al., 2019; Brown et al., 2020; Chowdhery et al., 2022). This progression in size has facilitated a shift from the conventional *pre-train and fine-tune* paradigm to *pre-train, prompt, and predict* paradigm (Kaplan et al., 2020; Liu et al., 2023). However, while prompting techniques have made remarkable progress in many areas of NLP, multi-step reasoning remained a challenging frontier where the scaling up appeared less effective (Rae et al., 2022).

The recent proposal of the *Chain-of-Thought* (CoT) *prompting* (Wei et al., 2022c), which prompts large language models (LLMs) to generate not only final answer but also intermediate reasoning steps, unleashed reasoning capabilities of LLMs. In response to this insight, zero-shot CoT (Kojima et al., 2022) shows CoT demonstrations can be replaced with a one-line instruction (e.g., "Let's think step by step.") and it significantly boosts zero-shot reasoning performance.

Although there has been notable progress in the field of multi-step reasoning in LLMs, there exists an untapped yet vital aspect of problem-solving. Human problem-solving is divided into two main phases in the information-processing theories of problem-solving in cognitive psychology: (a) construction of problem representation; (b) solution searching and its implementation (Newell, 1972; Greeno, 1978; Gick, 1986). When faced with a problem, humans first structurally encode the problem from two aspects, given information (initial state) and goal information (goal state) (Greeno, 1977; Gick, 1986). This structural interpretation is called problem representation and is the foundation of the human problem-solving process. Solution searching and subsequent implementation can be viewed as a series of cognitive processes that leads the initial state to the goal state (Davidson,

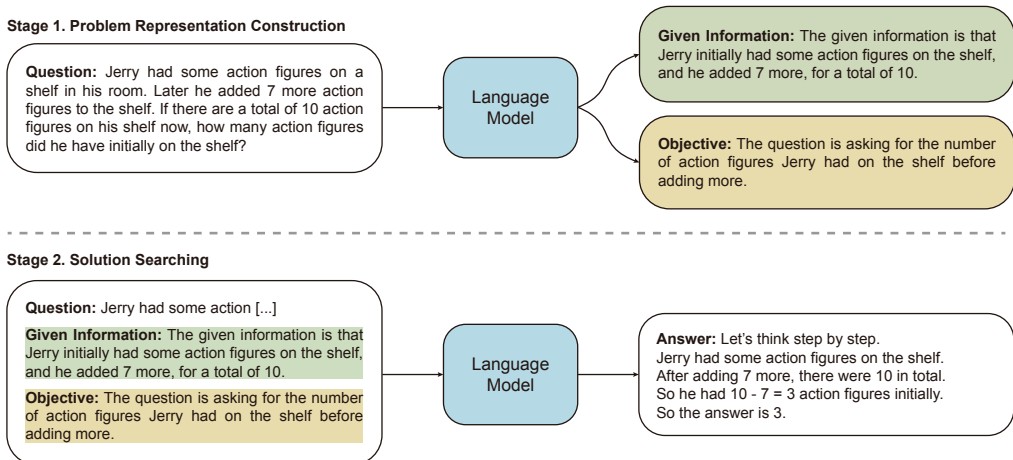

Figure 1: PRECoT consists of problem representation construction and solution searching stages. Integrating problem representation into solution searching process enhances reasoning in LLMs.

1994). However, studies in the reasoning of LLMs heavily focus on solution searching and its implementation. Those approaches include inducing LLMs to explicitly generate solution process without any control on its trajectory (Wei et al., 2022c; Kojima et al., 2022) and guiding LLMs to utilize specific solution strategies and procedures such as question decomposition (Fu et al., 2021; Zhou et al., 2023; Khot et al., 2023; Press et al., 2023), and planning (Wang et al., 2023; Hao et al., 2023; Yao et al., 2023). They initiate a reasoning process without any aid in understanding the problem. While problem representation construction is a crucial part of human problem-solving, as it contributes to solution searching or schema activation (Gick, 1986), it has yet to be explored whether this can also be beneficial to the reasoning in LLMs.

To investigate the effectiveness of problem representation in the reasoning of LLMs, we design **P**roblem **R**epresentation **E**nhanced **C**o**T** (PRECoT), a multi-stage prompting framework that helps LLMs build the problem representation. Similar to human problem representation, we consider the information specified in a problem as the initial state and the objective of the problem as the goal state. PRECoT sets these two states and incorporates them into the prompt to prime the solution process (Figure 1). We implement PRECoT in two fashions: few-shot and zero-shot prompting. Few-shot prompting leverages task demonstrations (Brown et al., 2020) to extract both states, while zero-shot prompting only utilizes instructions to obtain the states. We employ two transformer (Vaswani et al., 2017) based LLMs, PaLM 2 (Anil et al., 2023) and GPT-3 (Brown et al., 2020), to investigate the effectiveness of problem representation in the reasoning of LLMs. In extensive evaluation on a wide range of multi-step reasoning benchmarks, including arithmetic, commonsense, and symbolic reasoning, PRECoT outperforms few-shot and zero-shot CoT on most benchmarks. It validates the effectiveness of introducing problem representation in the reasoning of LLMs across such multiple domains. Additional analyses show notable strengths of problem representation grounded solution searching, such as robustness against irrelevant context and problem context sensitivity.

To sum up, our contribution is three-fold.

1. We propose a novel approach to improve reasoning in LLMs called PRECoT. It provides structured problem representation to LLMs and thereby enhances its solution process.

2. In extensive evaluation on multiple domain benchmarks (arithmetic, commonsense, and symbolic reasoning), our PRECoT outperforms both few-shot and zero-shot CoTs on most benchmarks, demonstrating its effectiveness. Additional analyses further support the benefit of problem representation.

3. Inspired by studies in cognitive psychology, this proposed approach offers useful perspectives into investigating LLMs' reasoning process.

## 2 RELATED WORK

**Large Language Model and Prompting.** Training transformer-based LMs at scale yields remarkable results on a wide range of NLP benchmarks (Devlin et al., 2019; Brown et al., 2020; Chowdhery et al., 2022). Scaled LMs (i.e., LLMs) show notable gradient-free learning capabilities with few-shot and zero-shot prompting (Brown et al., 2020; Kaplan et al., 2020; Liu et al., 2023; Wei et al., 2022b). In this paradigm, LLMs can generate responses to diverse tasks based on natural language descriptions without parameter updates. Few-shot prompting provides task demonstrations as a prompt for LLMs to generate a response similar to the examples in the demonstration. In zero-shot prompting, only a task instruction is given to LLMs to generate the desired response. In this study, we employ both prompting techniques to demonstrate the effectiveness of the proposed approach universally.

**Reasoning in Large Language Models.** CoT prompting (Wei et al., 2022c) exhibits that multi-step reasoning is one of the emergent abilities (Wei et al., 2022b) of LLMs. It induces the model to generate intermediate reasoning steps that lead to the final answer by priming LLMs with manually designed demonstrations. Kojima et al. (2022) substitutes the demonstration with a concise one-line instruction, "Let's think step by step." for the same effect. Numerous studies exploit this direction, including solution strategies such as problem decomposition (Fu et al., 2021; Khot et al., 2023; Zhou et al., 2023; Press et al., 2023) and planning (Yao et al., 2023; Wang et al., 2023). However, few works have focused on methods of representing the problem itself.

**Human Problem Solving.** The human problem-solving process has been modeled within information-processing theories of problem-solving in cognitive psychology (Polya, 1945; Newell, 1972; Gick, 1986; Bransford & Stein, 1993). Two processes are generally considered important: (a) generation of a problem representation, (b) solution searching and its implementation (Greeno, 1978; Simon, 1978). To solve a problem, humans first structure the problem from two perspectives, the given information (initial state) and the goal information (goal state), referred to as problem representation construction (Greeno, 1977). This representation allows the problem solver to search for a solution or invoke a schema activation (Gick, 1986). Solution searching and its implementation can be viewed as an active process that transforms the initial state into the goal state. This process can involve a variety of strategies, including solving by analogy, decomposition, planning, and means-end analysis (Polya, 1945; Simon, 1996; Gick, 1986). So far, in the reasoning in LLMs, only the solution searching process has been subject to exploration. To the best of our knowledge, our PRECOT is the first attempt to integrate problem representation into the reasoning process of LLMs, inspired by accumulated insights in cognitive psychology.

## 3 METHOD

As a cornerstone of the problem-solving process, humans first construct a problem representation, a structured interpretation of the problem that consists of the given information (initial state) and the goal information (goal state) (Greeno, 1977; Gick, 1986). To explore the implication of introducing this understanding process to reasoning in LLMs, we design a multi-stage prompting framework, PRECOT, and apply it to CoT prompting to reinforce it. PRECOT consists of two stages, *Problem Representation Construction* and *Solution Searching*. PRECOT builds problem representation in the earlier stage to help better generate reasoning steps in the subsequent stage (Figure 1). We implement PRECOT in both few-shot and zero-shot manners. Note that any training or fine-tuning is not required in our approach.

### 3.1 PROBLEM REPRESENTATION CONSTRUCTION

In this stage, every piece of information is collected from a given question and arranged to identify the initial state. In parallel, the objective (goal state) targeted by the question is refined in a concise form. To this end, the LLM is prompted with few-shot demonstrations (few-shot) or only instructions (zero-shot) to extract both states from the question. For the details of the prompts, please see Appendix A.1.

## 3.2 SOLUTION SEARCHING

After constructing the problem representation, we ask the LLM to solve the problem with the generated problem representation. As our main focus is to investigate the effectiveness of the problem representation in the reasoning of LLMs, we employ CoT prompting (Wei et al., 2022c; Kojima et al., 2022) as a solution searching method, which is simple and generic but powerful.

## 4 EXPERIMENTS

To evaluate the effectiveness of PRECoT, we conduct experiments with two different LLMs on benchmarks covering multiple reasoning domains. We compare our PRECoT with two existing approaches; Few-shot CoT (Wei et al., 2022c) and zero-shot CoT (Kojima et al., 2022). The results demonstrate that integrating problem representation into reasoning in LLMs makes notable performance improvement.

### 4.1 EXPERIMENTAL SETUP

**Benchmarks.** For comprehensive investigation, we use a wide range of tasks across three reasoning categories: (1) Arithmetic Reasoning, (2) Commonsense Reasoning, and (3) Symbolic Reasoning. These include 15 tasks from different sources, including Big-Bench Hard (Suzgun et al., 2023) that is a challenging subset of Big-Bench (Srivastava et al., 2023) (please see the sections in §4.2 for all the benchmarks we employ).

**Language Models.** To verify the generality of our approach while considering the CoT prompting tends to reliably work in LLMs (Wei et al., 2022b; Suzgun et al., 2023), we evaluate PRECoT using two Transformer (Vaswani et al., 2017) based large LMs with varying architectures, pre-training corpora and objectives, and instruction tuning methods. We only use the public engines of the models with greedy decoding and zero temperature for reproducibility.

- **PaLM 2**: We use a public engine of PaLM 2 (Anil et al., 2023), *text-bison-001*, through PaLM API. It is based on the scaled transformer architecture and pre-trained with various objectives following UL2 (Tay et al., 2023). Also, it is instruction-tuned with scaled FLAN procedure (Wei et al., 2022a; Chung et al., 2022; Longpre et al., 2023). Please note that the details of *text-bison-001* (e.g., exact model size) are not publicly available.
- **GPT-3**: We use a public engine of GPT-3 (Brown et al., 2020) with 175B parameters, *text-davinci-003*, through OpenAI API. It is a decoder-only auto-regressive transformer that inherits the architecture of GPT-2 (Radford et al., 2019). It is pre-trained with causal language modeling objective and instruction-tuned with reinforcement learning from human feedbacks (RLHF) (Christiano et al., 2017; Ouyang et al., 2022) using Proximal Policy Optimization (PPO) (Schulman et al., 2017).

**Baselines.** We compare our PRECoT with two representative approaches, few-shot and zero-shot CoT (Wei et al., 2022c; Kojima et al., 2022). For a fair comparison, we use the same chain of thought demonstrations as those used in previous works (Wei et al., 2022b; Suzgun et al., 2023) in the few-shot CoT and the solution searching stage of PRECoT. We manually annotate demonstrations to extract the given information and objective from the problem for few-shot PRECoT. We append "Let's think step by step" to the end of all few-shot prompts for generating reasoning steps to improve the performance (Kojima et al., 2022; Fu et al., 2023). Note that zero-shot PRECoT is an automated framework; any manually designed demonstration is not required, as in zero-shot CoT.

### 4.2 RESULTS

We observe the three main points in the results. (1) Our PRECoT outperforms CoT baselines on most benchmarks in both few-shot and zero-shot manners. (2) Our method enables qualitatively better reasoning in different aspects, such as robust reasoning against irrelevant information and context sensitivity. (3) Performance gains from applying our method are broadly consistent across both language models, demonstrating the generalizability of our approach. Please refer to the following sections for detailed evaluations on different domains.

Table 1: Accuracy (%) on arithmetic reasoning tasks of PRECoT compared to CoT (we highlight in green the cases where our method outperforms CoT). Few-shot PRECoT outperforms few-shot CoT on all tasks, and zero-shot PRECoT outperforms zero-shot CoT on most tasks.

| Model | Method | | GSM8K | GSM-IC | SVAMP | AQuA |
|---|---|---|---|---|---|---|
| PaLM 2 (N/A) | Few-shot | CoT | 56.56 | 60.50 | 74.30 | 42.91 |
| | | PRECoT | 61.79 (+5.23) | 74.05 (+13.55) | 78.20 (+3.90) | 44.49 (+1.58) |
| | Zero-shot | CoT | 56.86 | 61.90 | 72.30 | 41.73 |
| | | PRECoT | 54.74 (-2.12) | 67.85 (+5.95) | 73.90 (+1.60) | 45.67 (+3.94) |
| GPT-3 (175B) | Few-shot | CoT | 60.73 | 72.70 | 79.30 | 48.82 |
| | | PRECoT | 61.26 (+0.53) | 73.80 (+1.10) | 81.00 (+1.70) | 51.97 (+3.15) |
| | Zero-shot | CoT | 62.93 | 74.55 | 77.10 | 46.06 |
| | | PRECoT | 60.35 (-2.58) | 76.05 (+1.50) | 79.20 (+2.10) | 44.49 (-1.57) |

Table 2: Output comparison of few-shot PRECoT and CoT on GSM-IC in PaLM 2. While CoT gets distracted by irrelevant information and deviates from the correct reasoning trajectory, PRECoT keeps on track to reach the correct answer.

---

**Question:** A magazine costs $3 each. Jewel bought 10 magazines to be sold at $3.50 each. Jewel's mother bought 5 newspapers. How much will Jewel gain from selling her magazines? (Ground truth: 5)

---

| CoT (incorrect) | PRECoT (correct) |
|---|---|
| **Answer:** Let's think step by step. 10 magazines at $3 each is 10 x $3 = $30. Jewel will sell her magazines at $3.50 each so she will make 10 x $3.50 = $35. **Jewel's mother bought 5 newspapers at $1 each so she spent 5 x $1 = $5.** Jewel will gain $35 - $5 = $30 from selling her magazines. So the answer is **30**. | **Given Information:** The given information is 10 magazines were bought at $3 each, and will be sold at $3.50 each. Jewel's mother bought 5 newspapers.

**Objective:** The question is asking for the amount of money Jewel will gain from selling her magazines.

**Answer:** Let's think step by step. Jewel bought 10 magazines at $3 each, so that's 10 * 3 = $30. She will sell them at $3.50 each, so that's 10 * 3.5 = $35. So she will gain $35 - $30 = $5. So the answer is **5**. |

### 4.2.1 ARITHMETIC REASONING

**Tasks.** We consider the following four tasks for evaluation: (1) GSM8K (Cobbe et al., 2021), human-crafted grade school math word problems (MWPs) with linguistic diversity. (2) GSM-IC (Shi et al., 2023), a variant of GSM8K that introduces irrelevant information to the questions for investigating the distractibility of LMs. Since its size is around 60K, we use a random subset of size 2K for cost efficiency. (3) SVAMP (Patel et al., 2021), elementary-level MWPs designed to necessitate models to consider problem context instead of relying on simple heuristics. (4) AQuA-RAT (Ling et al., 2017), multiple-choice algebraic word problems augmented by crowdsourcing from a seed set including GRE (Graduate Record Examinations).

**Results.** Table 1 shows the accuracy of our method (PRECoT) and the competing methods (CoT). Overall, PRECoT boosts the performance of CoT on most tasks. Particularly, few-shot PRECoT makes significant improvements on all tasks over both LLMs. This result strongly suggests that problem representation is effective for arithmetic reasoning. Additionally, zero-shot PRECoT also shows competitive results on most tasks, but its effectiveness is slightly inferior to few-shot PRE-CoT. We speculate that this is because the problem representation generated in a few-shot setting is qualitatively better than that in a zero-shot setting (further analysis can be found in §5.2).

**Robustness to Irrelevant Information.** As demonstrated in the results on GSM-IC and SVAMP, PRECoT makes LLMs robust to disturbances from irrelevant information. In other words, PRECoT is more capable of finding the desired answer by combining only the necessary information from

Table 3: Accuracy (%) on commonsense reasoning tasks of PRECOT compared to COT. PRECOT outperforms COT on most tasks, demonstrating notable improvement in more complex tasks.

| Model | Method | | StrategyQA | CSQA | SocialIQA | Date | Causal Judg. | Ruin Names |
|---|---|---|---|---|---|---|---|---|
| PaLM 2 (N/A) | Few-shot | COT | 72.53 | 81.65 | 77.23 | 58.00 | 54.01 | 67.20 |
| | | PRECOT | 72.01 (-0.52) | 79.12 (-2.53) | 79.32 (+2.09) | 66.80 (+8.80) | 56.68 (+2.67) | 68.80 (+1.60) |
| | Zero-shot | COT | 77.16 | 78.30 | 75.18 | 55.60 | 58.29 | 60.40 |
| | | PRECOT | 73.32 (-3.84) | 80.59 (+2.29) | 77.07 (+1.89) | 58.00 (+2.40) | 60.43 (+2.14) | 62.00 (+1.60) |
| GPT-3 (175B) | Few-shot | COT | 73.06 | 78.54 | 73.80 | 81.20 | 62.03 | 76.40 |
| | | PRECOT | 71.97 (-1.09) | 79.52 (+0.98) | 74.05 (+0.25) | 84.40 (+3.20) | 62.57 (+0.54) | 77.60 (+1.20) |
| | Zero-shot | COT | 64.02 | 69.37 | 71.80 | 70.80 | 61.50 | 47.20 |
| | | PRECOT | 65.59 (+1.57) | 70.93 (+1.56) | 70.16 (-1.64) | 69.20 (-1.60) | 63.64 (+2.14) | 59.60 (+12.40) |

Table 4: Output comparison of few-shot PRECOT and COT on Date in PaLM 2 (it is originally a multiple-choice task, but the options are omitted here for the demonstration). While COT misses details of the context in the question, PRECOT is more sensitive to the details, enabling it to provide the correct answer.

| | |
|---|---|
| **Question:** Jane scheduled 3 appointments with 5 poeple for tomorrow (Tue, 7/9/1972). What is the date one week ago from today in MM/DD/YYYY? (Ground truth: 07/01/1972) | |
| **COT (incorrect)** | **PRECOT (correct)** |
| **Answer:** Let's think step by step. **7/9/1972 - 7 days = 7/2/1972.** So the answer is **07/02/1972**. | **Given Information:** The given information is Jane scheduled 3 apointments with 5 poeple for tomorrow (Tue, 7/9/1972). **Objective:** The question is asking for a date one week ago from today in MM/DD/YYYY. **Answer:** Let's think step by step. If tomorrow is 7/9/1972, then today is 7/8/1972. The date one week ago from today is 7/1/1972. So the answer is **07/01/1972**. |

distracting information. As depicted in Table 2, the problem representation enables the LLM to accurately trace the reasoning path from the initial state to the goal state without deviation.

### 4.2.2 COMMONSENSE REASONING

**Tasks.** We consider the following six tasks for evaluation: (1) StrategyQA (Geva et al., 2021), multi-hop open-domain yes-or-no questions. (2) CommonsenseQA (CSQA; Talmor et al. (2019)), multiple-choice questions that often require real-world prior knowledge. (3) SocialIQA (Sap et al., 2019), multiple-choice questions about human behavior and emotional responses in a given situation. (4) Date Understanding (Date), multiple-choice questions that require a model to calculate the date using simple date arithmetic and real-world knowledge. (5) Causal Judgment, multiple-choice questions that ask typical answers to causal questions related to a given context. (6) Ruin Names, multiple-choice questions that ask a humorous edit of the artist or movie name. Date, Causal Judgment, and Ruin Names are a subset of Big-Bench Hard.

**Results.** Table 3 shows the accuracy of our methods (PRECOT) and the competing methods (COT). Overall, PRECOT outperforms COT in few-shot and zero-shot manners over both LLMs. As one of the notable points, few-shot PRECOT makes substantial performance improvement for all commonsense reasoning tasks in both LLMs except for StrategyQA and CSQA. Unlike the other tasks, they only rely on internal prior knowledge. Therefore, we manually investigate the outputs of the LLMs and observe that constructing a good problem representation for these tasks is challenging (please see Appendix A.2 for an example). The evaluation result and this finding suggest that problem representation benefits complex commonsense reasoning tasks with a specific context, such as SocialIQA, Date, Causal Judgement, and Ruin Names.

Table 5: Accuracy (%) on symbolic reasoning tasks of PRECOT compared to COT. PRECOT out-performs COT on most tasks by a large margin.

| Model | Method | | Colors | Deduction | Tracking | Coin Flips | Last Letters |
|---|---|---|---|---|---|---|---|
| PaLM 2 (N/A) | Few-shot | COT | 70.00 | 47.07 | 38.40 | 70.27 | 44.53 |
| | | PRECOT | 74.80 (+4.80) | 64.27 (+17.20) | 42.93 (+4.53) | 96.40 (+26.13) | 43.86 (-0.67) |
| | Zero-shot | COT | 55.60 | 48.93 | 31.20 | 56.67 | 1.47 |
| | | PRECOT | 64.40 (+8.80) | 59.60 (+10.67) | 36.27 (+5.07) | 56.93 (+0.27) | 16.53 (+15.06) |
| GPT-3 (175B) | Few-shot | COT | 86.40 | 66.40 | 78.53 | 85.20 | 60.80 |
| | | PRECOT | 89.20 (+2.80) | 64.27 (-2.13) | 80.13 (+1.60) | 98.67 (+13.47) | 61.07 (+0.27) |
| | Zero-shot | COT | 69.20 | 54.40 | 36.80 | 98.40 | 51.07 |
| | | PRECOT | 76.80 (+7.60) | 60.53 (+6.13) | 36.40 (-0.40) | 92.27 (-6.13) | 61.07 (+10.00) |

Table 6: Output comparison of zero-shot PRECOT and COT on Colors in PaLM 2 (it is originally a multiple-choice task, but the options are omitted here for the demonstration). While PRECOT follows the correct goal-oriented reasoning trajectory, COT misunderstands the problem.

**Question:** On the nightstand, there are two green sheets of paper, two mauve cat toys, two green cat toys, and three mauve sheets of paper. If I remove all the cat toys from the nightstand, how many mauve things remain on it? (Ground truth: three)

| COT (incorrect) | PRECOT (correct) |
|---|---|
| **Answer:** Let's think step by step. 2 + 2 = **4 cat toys.** 2 + 3 = **5 mauve things. 5 - 4 = 1 mauve thing** remains on the nightstand. So the answer is **two**. | **Given Information:** The given information is 2 green sheets of paper, 2 mauve cat toys, 2 green cat toys, and 3 mauve sheets of paper on the nightstand. |
| | **Objective:** The question is asking how many mauve things remain on the nightstand after removing all the cat toys. |
| | **Answer:** Let's think step by step. 2 mauve cat toys + 3 mauve sheets of paper = **5 mauve things.** 5 mauve things - 2 mauve cat toys = **3 mauve things.** So the answer is **three**. |

**Context Sensitivity.** Our method shows relatively higher context awareness in commonsense reasoning. For example, as shown in Table 4, COT relies on simple heuristics to assemble information without regard to the context of the question, while PRECOT infers the appropriate answer based on the intent of the problem.

### 4.2.3 SYMBOLIC REASONING

**Tasks.** We consider the following five tasks for evaluation: (1) Reasoning about Colored Objects (Colors), multiple-choice questions about the consequence of taking a particular action on several portrayed objects. (2) Logical Deduction (Deduction), multiple-choice questions that ask for the order of objects based on the given relationship. (3) Tracking Shuffled Objects (Tracking), multiple-choice questions that ask for the final position of the object, given initial positions of objects and subsequent transformations (e.g., swaps). (4) Coin Flips, yes-or-no questions that ask for the final state of the coin after a few flips. (5) Last Letter Concatenation (Last Letters), questions that ask for the concatenation of the last letters from the given names. Colors, Deduction, and Tracking are a subset of Big-Bench Hard. Coin Flips and Last Letters are synthetically generated as described in Wei et al. (2022c).

**Results.** Table 5 shows the accuracy of our methods (PRECOT) and the competing methods (COT). Overall, our method significantly boosts the performance of both LLMs by a large margin on most tasks in both few-shot and zero-shot settings. Notably, improvements in Colors are consistently observed across all models and settings. Table 6 shows how setting the initial and goal states helps. The reasoning steps in COT are misaligned with the information embedded in the question, leading

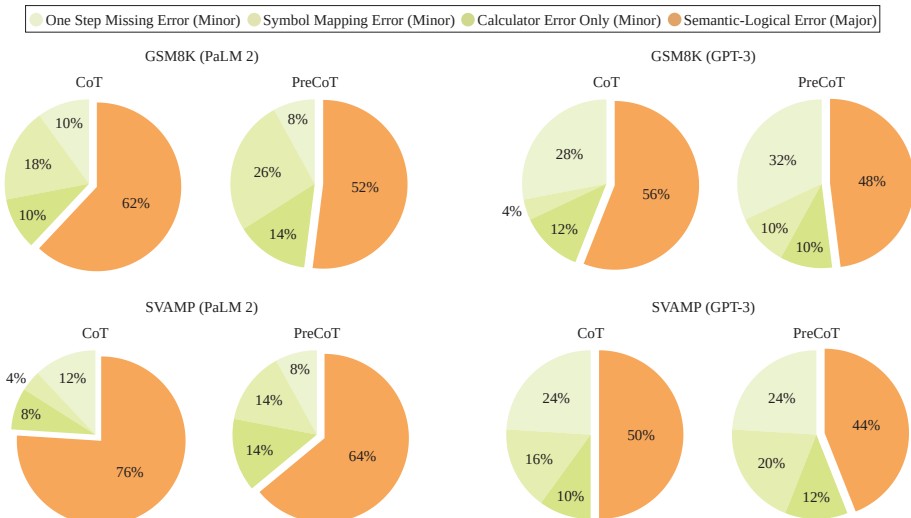

Figure 2: Distribution change in types of incorrect chain of thoughts evaluated on GSM8K and SVAMP in GPT-3 and PaLM 2. PRECoT reduces the major errors and makes it less severe.

them to deviate from the correct reasoning trajectory. In contrast, PRECoT demonstrates a proper reasoning process when presented with problem representation.

## 5 ANALYSIS

### 5.1 REASONING ERROR ANALYSIS

Evaluating the reasoning performance with accuracy might not be sufficient to measure the quality of the reasoning chains of LLMs when the generated answer is incorrect. Therefore, to explore the effectiveness of the problem representation in more detail, we conduct a manual analysis on the incorrect chain of thoughts on problems that a language model gets wrong. Specifically, for each LLM, we randomly sample 100 arithmetic reasoning problems (50 each from GSM8K and SVAMP), for which both CoT and PRECoT are incorrect. Then, we ask human evaluators to classify the types of incorrect chain of thoughts according to the criteria proposed by Wei et al. (2022c), which includes three minor error categories (one step missing, symbol mapping, and calculator error) and one major error category (semantic-logical error[1]). As shown in Figure 2, introducing problem representation reduces the major errors and makes them less severe errors in both LLMs (please see the example in Appendix A.2). The trend suggests that problem representation substantially contributes to the reasoning in LLMs. It is consistent with our performance evaluation presented in earlier sections, revealing the potential of problem representation that accuracy could not reflect.

### 5.2 PROBLEM REPRESENTATION QUALITY AND REASONING PERFORMANCE

In arithmetic reasoning tasks, we have observed cases in which poorly constructed problem representations lead to incorrect reasoning. Most of these issues arise from omissions while extracting given information, as shown in Table 7 (note that in many cases, the problem representation is well generated). Those tasks tend to have a more intricate context than other tasks, in which entities and their corresponding numeric values are heavily interweaved. This complexity poses challenges for zero-shot PRECoT in fully extracting the given information. This drawback could make it challenging to investigate the contribution of problem representation to reasoning in LLMs, which is our primary goal in this study. To determine if problem representation is effective on its own, independent of its construction process, we adjust the experiment protocol; we re-evaluate the solution searching performance of zero-shot PRECoT with the problem representation constructed in few-shot PRECoT.

---

[1] We merge two original major error categories, incoherent chain of thought and semantic understanding error, into semantic-logical error since they are closely related. Please see Appendix A.2 for the categorization detail.

Table 7: Output comparison of zero-shot PRECOT and zero-shot PRECOT+ on GSM8K in PaLM 2. Zero-shot PRECOT omits critical context when extracting the given information and fails to recognize the final objective clearly. On the contrary, zero-shot PRECOT+ can reach the correct answer, supported by the better-constructed problem presentation.

| **Question:** Jessica is trying to figure out how much to pay on all her debts each month. Her student loans have a minimum payment of $300/month, her credit card's minimum is $200/month, and her mortgage's minimum is $500/month. If Jessica wants to pay 50% more than the minimum, how much does she pay in a year? (Ground truth: 18000) | |
|---|---|
| **Zero-shot PRECOT (incorrect)** | **Zero-shot PRECOT+ (correct)** |
| **Given Information:** The given information is **300, 200, 500, 50%**.

**Objective:** The question is asking 1. how much Jessica pays on her debts each month 2. how much Jessica pays in a year.

**Answer:** Let's think step by step.
1. Jessica pays 300 + 200 + 500 = $1000/month.
2. Jessica pays 1000 * 12 = $12,000/year. Thus, the answer is **12000**. | **Given Information:** The given information is that Jessica has three debts with minimum payments of $300, $200, and $500, and she wants to pay 50% more than the minimum.

**Objective:** The question is asking how much Jessica pays in a year on all her debts.

**Answer:** Let's think step by step.
300 + 200 + 500 = $1000. 1000 * .5 = $500. 1000 + 500 = $1500. Jessica pays $1500 a month on her debts. 1500 * 12 = $18,000. Jessica pays $18,000 a year on her debts. Thus, the answer is **18000**. |

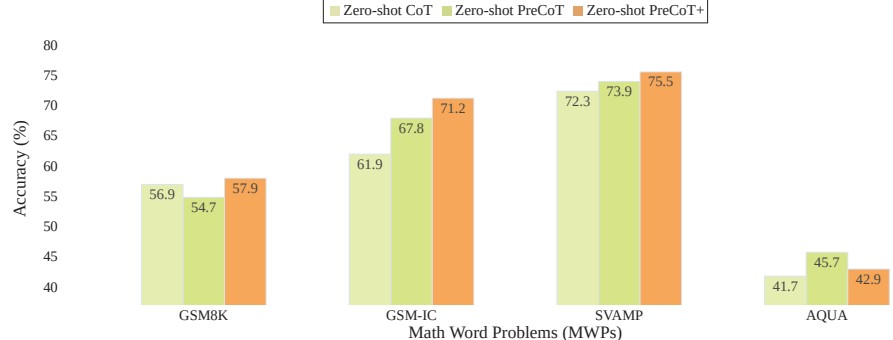

Figure 3: Zero-shot reasoning accuracy with problem representation generated with few-shot demonstrations (zero-shot PRECOT+). Zero-shot PRECOT+ tends to boost the performance of zero-shot PRECOT and outperforms zero-shot COT on all tasks.

We refer to this as zero-shot PRECOT+. Figure 3 shows that the problem representation generated in the few-shot setting further enhances the overall zero-shot reasoning performance. Notably, in GSM8K, our zero-shot PRECOT+ now outperforms zero-shot COT while it does not when problem representation is constructed in a zero-shot manner. These findings suggest that a better-constructed problem representation can more effectively contribute to the reasoning process.

## 6 CONCLUSION

We propose a novel approach called PRECOT that enhances the reasoning in LLMs by incorporating problem representation, a cornerstone in human problem-solving. The extensive evaluation on 15 benchmarks across three reasoning categories demonstrates PRECOT outperforms COT on most tasks in both few-shot and zero-shot fashions. Furthermore, a qualitative error analysis indicates that PRECOT reduces major reasoning errors. Additionally, the improvement in zero-shot PRECOT+ shows that well-constructed problem representations enhances reasoning performance, implying that introducing problem representations is an effective approach. We hope our insights will inspire future work exploring the potential of language models for reasoning.

REPRODUCIBILITY STATEMENT

For reproducibility, we attach our implementation of PRECOT, evaluation code bases, and all prompts as supplementary materials. Our results can be reproduced using the attached codes and aforementioned public APIs.

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

# A  APPENDIX

## A.1  PRECOT IMPLEMENTATION DETAIL

**Problem Representation Construction.**   We use *task demonstrations* (few-shot PRECOT) or *instructions* (zero-shot PRECOT) with generation *triggers* to extract the given information and objective from a problem. We format the prompt as shown in Table 8 with its components in Table 10.

**Solution Searching.**   We employ few-shot and zero-shot CoT prompting (Wei et al., 2022c; Kojima et al., 2022) for generating reasoning steps. We format the prompts as shown in Table 9. Note that the task demonstrations are only used in few-shot PRECOT. We extract LLMs' predictions (predicted label) from generated reasoning steps using a manually designed rule-based output cleanser. We attach evaluation code bases in the supplementary material. Only for zero-shot PRECOT, we additionally extract their predictions with LLMs and then apply the output cleanser following Kojima et al. (2022).

Table 8: Prompt format for generating problem representation in PRECOT.

```
{A task demonstration or an instruction}

Question: {Question}
{Trigger}
```

Table 9: Prompt format for generating reasoning steps in PRECOT.

```
{A task demonstration (few-shot PRECOT only)}

Question: {Question}
Given Information: {Extracted information in the previous stage.}
Objective: {Extracted objective in the previous stage.}
Answer: Let's think step by step.
```

## A.2  ADDITIONAL ERROR ANALYSIS

**Arithmetic Reasoning.**   We manually investigate the incorrect chain of thoughts generated by both LLMs within few-shot CoT and PRECOT on GSM8K and SVAMP. We largely follow the criteria proposed by Wei et al. (2022c) for a fair comparison, but we slightly modify the major error categorization. Specifically, we merge two original major error categories, incoherent chain of thought and semantic understanding error, into semantic-logical error. This is because these two categories are not independent, which makes it very difficult to distinguish between them. The concise definitions of each error category are as follows. For a detailed description of these criteria and examples, please refer to the previous work (Wei et al., 2022c).

- **Calculator Error Only (minor)**: In this category, attaining the correct answer is possible by rectifying all calculation errors present in the reasoning chain.
- **Symbol Mapping Error (minor)**: In this category, rectifying the numeric symbols in the equation without modifying the words in the reasoning chain makes an utterly correct explanation.
- **One Step Missing Error (minor)**: In this category, the path to the correct answer necessitates the inclusion of one logical step in the reasoning chain.
- **Semantic-Logical Error (major)**: Unlike the minor errors above, this category represents a scenario where the reasoning chain contains severe semantic and logical flaws, making the attainment of the correct answer impossible through simple modifications.

As delineated in the analysis section, PRECOT successfully downgrades some major errors in the incorrect chain of thoughts to minor ones. In the example in Table 11, CoT exhibits a significant logical flaw, whereas PRECOT only makes a computational mistake.

Table 10: Details of the components that comprise the prompt for constructing the problem representation in PRECOT.

| Task | Component | Content |
|---|---|---|
| Given Information Extraction | Task demonstration (few-shot PRECOT) | Please refer §A.4 to see full prompt for each tasks. |
| | Instruction (zero-shot PRECOT) | `Extract what information is given in the following question to help answer it.` |
| | Trigger | `Given Information: The given information is` |
| Objective Extraction | Task demonstration (few-shot PRECOT) | Please refer §A.4 to see full prompt for each tasks. |
| | Instruction (zero-shot PRECOT) | `Write what the following question is asking for to help answer it.` |
| | Trigger | `Objective: The question is asking` |

Table 11: Output comparison of few-shot PRECOT and CoT on GSM8K in PaLM 2. PRECOT only exhibit a minor calculation error, while CoT has a logical flaw and misses a critical reasoning step.

**Question:** Hannah slips on a banana peel and breaks her arm. The doctor charges her $200 for the cast, $300/hour for a 30-minute visit, $4/pill for 30 painkillers, and $6/hour for a mandatory minimum of 2 hours of parking. How much does the doctor's visit cost total? (Ground truth: 482)

| CoT (Semantic-Logical Error) | PRECOT (Calculation Error) |
|---|---|
| **Answer:** Let's think step by step. 300 / 60 = 5 dollars **per 30 minutes**. **5 \* 30 = $150** for the visit. 30 \* 4 = $120 for the painkillers. **150 + 120 + 200 = $470**. So the answer is 470. | **Given Information:** The given information is that Hannah's doctor charges $200 for the cast, $300/hour for a 30-minute visit, $4/pill for 30 painkillers, and $6/hour for a mandatory minimum of 2 hours of parking. **Objective:** The question is asking for the total cost of the doctor's visit. **Answer:** Let's think step by step. The cast costs $200. The visit costs 300 / 60 = $5 / minute. The visit costs 5 \* 30 = $150. The painkillers cost 4 \* 30 = $120. The parking costs 6 \* 2 = $12. The total cost is **200 + 150 + 120 + 12 = $582**. So the answer is **582**. |

**Commonsense Reasoning.** Tasks like StrategyQA and CSQA, which have no in-question information to reference and only rely on internal prior knowledge, are less likely to have well-constructed problem representations. These problem representations are often observed in the cases where the language model reaches incorrect answers, suggesting that problem representation might be less helpful to this type of question, as shown in Table 12.

## A.3 DATASET DETAILS

We list the following 15 benchmarks we use in the experiments to aid reproducibility. Coin Flips and Last Letters are synthetically generated as described in previous works (Wei et al., 2022c; Kojima et al., 2022). Since the size of GSM-IC is about 60K, we use random subsets of size 2K for cost efficiency.

## A.4 PROMPTS

We provide all prompts we use in the main experiment to aid reproducibility. In the solution searching stage of few-shot PRECOT, we use the same chain of thoughts as those used in previous

Table 12: Output example of zero-shot PRECOT on StrategyQA in PaLM 2. StrategyQA has no information to extract and relies solely on internal prior knowledge, making it hard for PRECOT to construct problem representation.

| | |
|---|---|
| **Question:** Can vitamin C rich fruits be bad for health? (Ground truth: yes) | |
| **PRECOT (incorrect)** | |
| **Given Information:** The given information is 'vitamin C rich fruits'. **Objective:** The question is asking whether vitamin C rich fruits can be bad for health. **Answer:** Let's think step by step. Vitamin C rich fruits are good for health. So the final answer is no. | |

Table 13: Details about the benchmark tasks employed in our experiments.

| Category | Dataset | Size | Split |
|---|---|---|---|
| Arithmetic Reasoning | GSM8K (Cobbe et al., 2021) | 1,319 | test |
| | GSM-IC (Shi et al., 2023) | 2,000 | 2/m-step |
| | SVAMP (Patel et al., 2021) | 1,000 | test |
| | AQuA (Ling et al., 2017) | 254 | test |
| Commonsense Reasoning | CommonsenseQA (Talmor et al., 2019) | 1,221 | dev (rand) |
| | StrategyQA (Geva et al., 2021) | 2,289 | Big-Bench (Srivastava et al., 2023) |
| | SocialIQA (Sap et al., 2019) | 1,953 | dev |
| | Date Understanding (Srivastava et al., 2023) | 250 | Big-Bench Hard (Suzgun et al., 2023) |
| | Causal Judgement (Srivastava et al., 2023) | 250 | |
| | Ruin Names (Srivastava et al., 2023) | 250 | |
| Symbolic Reasoning | Reasoning about Colored Objects (Srivastava et al., 2023) | 250 | |
| | Logical Deduction (Srivastava et al., 2023) | 750 | |
| | Tracking Shuffled Objects (Srivastava et al., 2023) | 750 | |
| | Coin Flips (Wei et al., 2022c) | 750 | 3/5/7 |
| | Last Letters (Wei et al., 2022c) | 750 | 3/5/7 |

works (Wei et al., 2022c; Suzgun et al., 2023) for a fair comparison[2]. In this paper, we only attach the prompts we use for arithmetic reasoning tasks, including GSM8K, GSM-IC, and SVAMP[3]. Please see Table 14- 21 and refer to the supplementary material for all other prompts.

---

[2]SocialIQA is an exception, as there is no CoT prompt available. Therefore, we manually annotated few-shot prompts, including the solution searching stage.

[3]For AQuA-RAT, please refer to the supplementary materials.

Table 14: Prompt for generating reasoning steps in zero-shot CoT.

Question: {Question}
Answer: Let's think step by step.

Table 15: Prompt for extracting given information from a question in zero-shot PRECoT.

Extract what information is given in the following question to help answer it.

Question: {Question}
Given Information: The given information is

Table 16: Prompt for extracting an objective from a question in zero-shot PRECoT.

Write what the following question is asking for to help answer it.

Question: {Question}
Objective: The question is asking

Table 17: Prompt for generating reasoning steps in zero-shot PRECoT.

Question: {Question}
Given Information: {Extracted information in the previous stage.}
Objective: {Extracted objective in the previous stage.}
Answer: Let's think step by step.

Table 18: Prompt for generating reasoning steps in arithmetic reasoning tasks in few-shot CoT. We use the same questions and the chain of thoughts as Wei et al. (2022c) for a fair comparison. Also, to improve performance, we add "`Let's think step by step`" to the beginning of the chain of thoughts.

---

**Question:** There are 15 trees in the grove. Grove workers will plant trees in the grove today. After they are done, there will be 21 trees. How many trees will the grove workers plant today?
**Answer:** Let's think step by step.
There are 15 trees originally.
Then there were 21 trees after some more were planted.
So there must have been 21 - 15 = 6.
So the answer is 6.

**Question:** If there are 3 cars in the parking lot and 2 more cars arrive, how many cars are in the parking lot?
**Answer:** Let's think step by step.
There are originally 3 cars.
2 more cars arrive.
3 + 2 = 5.
So the answer is 5.

**Question:** Leah had 32 chocolates and her sister had 42. If they ate 35, how many pieces do they have left in total?
**Answer:** Let's think step by step.
Originally, Leah had 32 chocolates.
Her sister had 42.
So in total they had 32 + 42 = 74.
After eating 35, they had 74 - 35 = 39.
So the answer is 39.

**Question:** Jason had 20 lollipops. He gave Denny some lollipops. Now Jason has 12 lollipops. How many lollipops did Jason give to Denny?
**Answer:** Let's think step by step.
Jason started with 20 lollipops.
Then he had 12 after giving some to Denny.
So he gave Denny 20 - 12 = 8.
So the answer is 8.

**Question:** Shawn has five toys. For Christmas, he got two toys each from his mom and dad. How many toys does he have now?
**Answer:** Let's think step by step.
Shawn started with 5 toys.
If he got 2 toys each from his mom and dad, then that is 4 more toys.
5 + 4 = 9.
So the answer is 9.

**Question:** There were nine computers in the server room. Five more computers were installed each day, from monday to thursday. How many computers are now in the server room?
**Answer:** Let's think step by step.
There were originally 9 computers.
For each of 4 days, 5 more computers were added.
So 5 * 4 = 20 computers were added.
9 + 20 is 29.
So the answer is 29.

**Question:** Michael had 58 golf balls. On tuesday, he lost 23 golf balls. On wednesday, he lost 2 more. How many golf balls did he have at the end of wednesday?
**Answer:** Let's think step by step
Michael started with 58 golf balls.
After losing 23 on tuesday, he had 58 - 23 = 35.
After losing 2 more, he had 35 - 2 = 33 golf balls.
So the answer is 33.

**Question:** Olivia has $23. She bought five bagels for $3 each. How much money does she have left?
**Answer:** Let's think step by step.
Olivia had 23 dollars.
5 bagels for 3 dollars each will be 5 x 3 = 15 dollars.
So she has 23 - 15 dollars left.
23 - 15 is 8.
So the answer is 8.

---

Table 19: Prompt for extracting given information from the problems in arithmetic reasoning tasks in few-shot PRECOT.

---

**Question:** There are 15 trees in the grove. Grove workers will plant trees in the grove today. After they are done, there will be 21 trees. How many trees will the grove workers plant today?
**Given Information:** The given information is that the workers will add trees to the grove today. There were originally 15 trees in the grove, and there should be 21 after they are done.

**Question:** If there are 3 cars in the parking lot and 2 more cars arrive, how many cars are in the parking lot?
**Given Information:** The given information is that two more cars arrive at the parking lot with three cars.

**Question:** Leah had 32 chocolates and her sister had 42. If they ate 35, how many pieces do they have left in total?
**Given Information:** The given information is that Leah and her sister initially had 32 and 42 chocolates, respectively, and together ate 35.

**Question:** Jason had 20 lollipops. He gave Denny some lollipops. Now Jason has 12 lollipops. How many lollipops did Jason give to Denny?
**Given Information:** The given information is that Jason initially had 20 lollipops, and after giving some to Denny, it was reduced to 12.

**Question:** Shawn has five toys. For Christmas, he got two toys each from his mom and dad. How many toys does he have now?
**Given Information:** The given information is that Shawn initially had 5 toys and received 2 from each of his parents.

**Question:** There were nine computers in the server room. Five more computers were installed each day, from monday to thursday. How many computers are now in the server room?
**Given Information:** The given information is that the server room with nine computers had five additional computers installed daily, Monday through Thursday.

**Question:** Michael had 58 golf balls. On tuesday, he lost 23 golf balls. On wednesday, he lost 2 more. How many golf balls did he have at the end of wednesday?
**Given Information:** The given information is that Michael initially had 57 golf balls, but he lost 23 on Tuesday and two more on Wednesday.

**Question:** Olivia has $23. She bought five bagels for $3 each. How much money does she have left?
**Given Information:** The given information is that Olivia initially had $23, and she bought five bagels for $3 each.

---

Table 20: Prompt for extracting objective from the problems in arithmetic reasoning tasks in few-shot PRECOT.

---

**Question:** There are 15 trees in the grove. Grove workers will plant trees in the grove today. After they are done, there will be 21 trees. How many trees will the grove workers plant today?
**Objective:** The question is asking for the number of trees the workers must add to the grove today.

**Question:** If there are 3 cars in the parking lot and 2 more cars arrive, how many cars are in the parking lot?
**Objective:** The question is asking for the number of cars will be in the parking lot.

**Question:** Leah had 32 chocolates and her sister had 42. If they ate 35, how many pieces do they have left in total?
**Objective:** The question is asking for the number of chocolates left after they eat.

**Question:** Jason had 20 lollipops. He gave Denny some lollipops. Now Jason has 12 lollipops. How many lollipops did Jason give to Denny?
**Objective:** The question is asking for the number of lollipops Jason gave to Denny.

**Question:** Shawn has five toys. For Christmas, he got two toys each from his mom and dad. How many toys does he have now?
**Objective:** The question is asking for the number of toys Shawn owns after receiving the gift.

**Question:** There were nine computers in the server room. Five more computers were installed each day, from monday to thursday. How many computers are now in the server room?
**Objective:** The question is asking for the total number of computers in the server room after installing additional computers.

**Question:** Michael had 58 golf balls. On tuesday, he lost 23 golf balls. On wednesday, he lost 2 more. How many golf balls did he have at the end of wednesday?
**Objective:** The question is asking for the number of balls Michael has at the end of Wednesday.

**Question:** Olivia has $23. She bought five bagels for $3 each. How much money does she have left?
**Objective:** The question is asking how much money Olivia has left after buying bagels.

---

Table 21: Prompt for generating reasoning steps for arithmetic reasoning tasks in few-shot PRECoT. This prompt is a combination of the three prompts presented earlier (Table 18, 19, 20).

---

**Question:** There are 15 trees in the grove. Grove workers will plant trees in the grove today. After they are done, there will be 21 trees. How many trees will the grove workers plant today?
**Given Information:** The given information is that the workers will add trees to the grove today. There were originally 15 trees in the grove, and there should be 21 after they are done.
**Objective:** The question is asking for the number of trees the workers must add to the grove today.
**Answer:** Let's think step by step.
There are 15 trees originally.
Then there were 21 trees after some more were planted.
So there must have been 21 - 15 = 6.
So the answer is 6.

**Question:** If there are 3 cars in the parking lot and 2 more cars arrive, how many cars are in the parking lot?
**Given Information:** The given information is that two more cars arrive at the parking lot with three cars.
**Objective:** The question is asking for the number of cars will be in the parking lot.
**Answer:** Let's think step by step.
There are originally 3 cars.
2 more cars arrive.
3 + 2 = 5.
So the answer is 5.

**Question:** Leah had 32 chocolates and her sister had 42. If they ate 35, how many pieces do they have left in total?
**Given Information:** The given information is that Leah and her sister initially had 32 and 42 chocolates, respectively, and together ate 35.
**Objective:** The question is asking for the number of chocolates left after they eat.
**Answer:** Let's think step by step.
Originally, Leah had 32 chocolates.
Her sister had 42.
So in total they had 32 + 42 = 74.
After eating 35, they had 74 - 35 = 39.
So the answer is 39.

**Question:** Jason had 20 lollipops. He gave Denny some lollipops. Now Jason has 12 lollipops. How many lollipops did Jason give to Denny?
**Given Information:** The given information is that Jason initially had 20 lollipops, and after giving some to Denny, it was reduced to 12.
**Objective:** The question is asking for the number of lollipops Jason gave to Denny.
**Answer:** Let's think step by step.
Jason started with 20 lollipops.
Then he had 12 after giving some to Denny.
So he gave Denny 20 - 12 = 8.
So the answer is 8.

**Question:** Shawn has five toys. For Christmas, he got two toys each from his mom and dad. How many toys does he have now?
**Given Information:** The given information is that Shawn initially had 5 toys and received 2 from each of his parents.
**Objective:** The question is asking for the number of toys Shawn owns after receiving the gift.
**Answer:** Let's think step by step.
Shawn started with 5 toys.
If he got 2 toys each from his mom and dad, then that is 4 more toys.
5 + 4 = 9.
So the answer is 9.

**Question:** There were nine computers in the server room. Five more computers were installed each day, from monday to thursday. How many computers are now in the server room?
**Given Information:** The given information is that the server room with nine computers had five additional computers installed daily, Monday through Thursday.
**Objective:** The question is asking for the total number of computers in the server room after installing additional computers.
**Answer:** Let's think step by step.
There were originally 9 computers.
For each of 4 days, 5 more computers were added.
So 5 * 4 = 20 computers were added.
9 + 20 is 29.
So the answer is 29.

**Question:** Michael had 58 golf balls. On tuesday, he lost 23 golf balls. On wednesday, he lost 2 more. How many golf balls did he have at the end of wednesday?
**Given Information:** The given information is that Michael initially had 57 golf balls, but he lost 23 on Tuesday and two more on Wednesday.
**Objective:** The question is asking for the number of balls Michael has at the end of Wednesday.
**Answer:** Let's think step by step.
Michael started with 58 golf balls.
After losing 23 on tuesday, he had 58 - 23 = 35.
After losing 2 more, he had 35 - 2 = 33 golf balls.
So the answer is 33.

**Question:** Olivia has $23. She bought five bagels for $3 each. How much money does she have left?
**Given Information:** The given information is that Olivia initially had $23, and she bought five bagels for $3 each.
**Objective:** The question is asking how much money Olivia has left after buying bagels.
**Answer:** Let's think step by step.
Olivia had 23 dollars.
5 bagels for 3 dollars each will be 5 x 3 = 15 dollars.
So she has 23 - 15 dollars left.
23 - 15 is 8.
So the answer is 8.

---

