# OpenReview forum: "PreCoT: Problem Representation Enhances Reasoning in Large Language Models"
_ICLR.cc/2024/Conference — Submitted to ICLR 2024_

### Official Review · Reviewer_2rHa · 2023-10-24

**Soundness:** 3 good
**Presentation:** 3 good
**Contribution:** 3 good
**Rating:** 8
**Confidence:** 5

**Summary:**

This paper is a version of Chain of Thought (CoT) methods for LLMs that improves performance by explicitly including problem representation as well as goal state and method. They illustrated this improvement using three benchmark datasets of reasoning tasks.

**Strengths:**

The strongest aspect of this paper was the experimental section because of broad application of techniques. The datasets they used were standard and well suited to the tasks at hand as well as providing a clear baseline.

I additionally appreciated the clear examples interspersed through the paper and the detailed appendix.

Finally, I thought the analysis section was well written and thoughtful.

In terms of the key dimensions:

- Originality: I thought the original part of this paper was using the psychological notion of goal and solution state to improve LLMs. The actual solution of "a better prompt" did show promise but felt to me more like a variation on a theme of existing methods than on anything substantially new.

- Quality: Results were well baselined and quantified.

- Clarity: This was well written, with a clear setup and motivation.

- Significance: Incorporation and understanding of abstract reasoning with LLMs is a major challenge and any improvements there are significant from the perspective of improved LLM performance as well as from the perspective of explainability and generalizability.

**Weaknesses:**

The first weakest part of this paper in my mind was the methods section, which seemed to have less details than the other parts of the paper.  I was specifically looking for more details on the few-shot and zero-shot methods. I also thought that the robustness claims made in the experimental section needed to be introduced as part of the methods.

Perhaps more significant weakness in this paper is that the performance results didn't improve as much as I would expect based on the stated hypothesis that key parts of problem are being left out of the problem formulation. I'm not sure that is a fundamental weakness and might be addressable in the discussion or the introduction.

The specific LLMs used are not state of the art, so there is also a question of if these methods make as big a difference with the most recent generation of generative models.

**Questions:**

I really appreciated the robustness results in the experimental section, but one of the questions I had after reading the paper is about robustness and stability of the prompting method. It is not difficult to make LLMs change their response based on the prompt, and I don't think this work is sufficient to to claim that the performance improvements are attributable to the specifics of the prompt used rather than the natural variation due to changes in prompts more generally.

Also, I believe the references are ill-formated - page 13 and 14 are a single paper. Can you check that those are correct?

---

> ### Author Response · Authors · 2023-11-17
> **Response to Reviewer 2rHa (1/2)**
>
> We appreciate your detailed review and constructive feedback. Also, thank you for highlighting our paper's clarity, quality, experimental design, and significance. In response to your comments, we've added more explanations and extra experimental data in the following sections to address your points. We would like to hear any further feedback, suggestions, or questions you might have.
>
> ---
>
> **(1) Variation on a theme of existing methods than on anything substantially new.**
>
> Thank you for the feedback. While we respect your opinion, we would like to highlight the key distinction between existing methods and ours.
>
> The core of our idea is not to extract some information from a question but to ground solution searching with the interpretation of a problem observed in human problem-solving. As described in the introduction section, only the solution-searching method has been subject to exploration in this field.
>
> Our method can easily and universally be adapted to other solution searching methods in a plug-and-play manner. To show this, as suggested by Reviewer 9iGg, we implemented Problem Representation Enhanced Least-to-Most prompting (PreLtM) along with vanilla LtM [1]. When comparing PreLtM to LtM, the superiority of PreLtM highlights our approach's general applicability and effectiveness in solution searching.
>
> |  | LtM | PreLtM |
> | --- | --- | --- |
> | GSM8K | 54.51 | 57.09 (+2.58) |
> | GSM-IC | 62.50 | 73.40 (+10.90) |
> | DROP (football) | 71.60 | 72.80 (+1.20) |
> | DROP (non-football) | 78.80 | 79.80 (+1.00) |
> | Last Letters | 50.40 | 48.53 (-1.87) |
>
>
> **(2) Details in method section.**
>
> We will update the methods section for better clarity. Please understand that we are unable to revise the paper immediately, as additional experiments were conducted during the discussion session, and we need to redistribute the space to include the results and implications of these. For this round, we improved the visibility of Figure 1 (overview of our method). Also, please note that we have provided a detailed implementation of our method and prompts in the appendix section.
>
> Additionally, we would like to emphasize our commitment to open science. We attached all prompts and evaluation code bases we used in the supplementary material to ensure clarity, reproducibility, and fairness.
>
> **(3) Performance improvement.**
>
> Firstly, we would like to emphasize some notable improvements. PreCoT and zero-shot PreCoT+ surpass CoT on all arithmetic reasoning benchmarks. For other reasoning tasks, empirical results show that problem representation consistently boosts CoT on most tasks in both few-shot and zero-shot settings. These results include notable improvements on GSM8K (+5.23), GSM-IC (+13.55), Date Understanding (+8.80), Colors (+4.80), Logical Deduction (+17.20), and Coin Flips (+26.13) in few-shot setting in PaLM 2. Moreover, problem representation significantly reduces major reasoning errors and makes them less severe in both LLMs (-11% in PaLM 2, -7% in GPT-3 on average) (Please refer to Section 5.1.). Note that these improvements are achieved without any task-specific solution step manipulation.
>
> Furthermore, when we employ LtM as solution searching method instead of CoT, the results on PreLtM show the general applicability and effectiveness of our approach with other solution searching method. We observe consistent improvements on all arithmetic reasoning tasks, highlighting the general effectiveness of problem representation.
>
> **(4) The specific LLMs used are not state of the art.**
>
> We used the state-of-the-art models available to us, considering the scale of our evaluation. PaLM 2 is the most recent LLM proposed by Google, and text-davinci-003 is also one of the most recent models among the GPT-3 splits. We employ the major LLMs from different groups (with varying architectures, training corpora, and training objectives) to rigorously validate the general effectiveness of our method. These models are actively used in this domain. GPT-4 could not be used for our evaluation as it would entail prohibitive costs (both in terms of time and money) for our extensive evaluation. Instead, we tried rigorously validating our method with various benchmarks using the latest, cost-effective models.

---

> ### Author Response · Authors · 2023-11-17
> **Response to Reviewer 2rHa (2/2)**
>
> **(5) Robustness of our method.**
>
> In the problem representation construction stage, a unique problem representation is generated by LLM for each problem sample. Therefore, the problem representation that primes the solution searching process is different for each problem. So, specific input manipulations could not be applied consistently in both few-shot and zero-shot settings.
>
> In this case, a consistent trend of performance improvement is observed across few-shot PreCoT, zero-shot PreCoT, and zero-shot PreCoT+. This suggests that such improvements are not based on simple input perturbations but rather on the robust effectiveness of our approach.
>
> Furthermore, we observe consistent performance improvements even if we change the solution searching method with LtM. It also confirms the robustness of our approach.
>
> Additionally, when comparing the performance of our method with variants replacing the problem representation with question repetition (as suggested by Reviewer c8na), our consistent superiority demonstrates that the performance improvement from problem representation is not grounded on information repetition (R1: question repetition once; R2: question repetition twice).
>
> |  | R1 + CoT | PreCoT |
> | --- | --- | --- |
> | GSM8K | 57.62 | 61.79 (+4.17) |
> | GSM-IC | 65.35 | 75.50 (+10.15) |
> | SVAMP | 74.00 | 78.20 (+4.20) |
> | AQuA | 43.70 | 44.49 (+0.79) |
>
> |  | R2 + CoT | PreCoT |
> | --- | --- | --- |
> | GSM8K | 55.72 | 61.79 (+6.07) |
> | GSM-IC | 63.50 | 75.50 (+12.00) |
> | SVAMP | 74.00 | 78.20 (+4.20) |
> | AQuA | 38.58 | 44.49 (+5.91) |
>
> We will update our paper reflecting your thoughtful question after receiving further feedback you may have.
>
>
> **(6) Single reference across multiple pages.**
>
> The paper you mentioned is a collaborative work contributed by 450 authors across 132 institutions. We did not truncate the author list to express our respect to their open-source contribution in investigating the capabilities of LLMs.
>
> ---
>
> **References**
>
> [1] Zhou, D., Schärli, N., Hou, L., Wei, J., Scales, N., Wang, X., ... & Chi, E. H. (2022, September). Least-to-Most Prompting Enables Complex Reasoning in Large Language Models. In The Eleventh International Conference on Learning Representations.

---

> ### Comment · Reviewer_2rHa · 2023-11-21
>
> Thank you, authors, for all the useful discussion and thoughtful answers to my questions. Based on your comments I have updated my review, both in terms of confidence and paper quality. I'm excited to see this kind of work being done and look forward to more like it in the future.

---

> > ### Author Response · Authors · 2023-11-22
> >
> > We greatly appreciate your time in reviewing our paper and are delighted with your constructive and encouraging feedback. We will update the paper based on your suggestions.

---

### Official Review · Reviewer_9iGg · 2023-10-25

**Soundness:** 3 good
**Presentation:** 3 good
**Contribution:** 2 fair
**Rating:** 5
**Confidence:** 5

**Summary:**

This paper integrates the construction of problem representation into an LLM's problem-solving process to enhance its reasoning capability. The construction of problem representation involves defining the initial and goal states of a problem. The PreCoT method comprises two key stages: 1) In the first stage, when provided with a problem statement, it guides the LLM to extract the given information and the objective. 2) In the second stage, it appends the extracted information and objective to the problem statement, and employs zero-shot or few-shot CoT prompting to guide the LLM in its reasoning process. The experiments involving arithmetic, commonsense, and symbolic reasoning showcase that PreCoT achieves higher accuracy than CoT in both zero-shot and few-shot settings in most cases. Furthermore, additional analyses reveal that PreCoT exhibits greater resilience to irrelevant information in arithmetic reasoning and demonstrates relatively higher contextual awareness in commonsense reasoning.

Contributions: 1) This paper underscores the significance of "problem representation construction" as a preliminary step before initiating the reasoning process within LLM. It entails the extraction of the given information and problem objective to form the problem representation. 2) Empirical results provide concrete evidence of the method's effectiveness.

**Strengths:**

(1) The method is simple yet effective, consistently outperforming the baseline CoT in the majority of cases.

(2) The study is supported by a wealth of comprehensive experiments.

**Weaknesses:**

(1） Its technical novelty is somewhat limited, as the idea of initially guiding LLM to extract given information or objectives lacks novelty. In two of the previous works https://aclanthology.org/2023.acl-long.147.pdf and https://arxiv.org/abs/2306.03872, they also involve extracting the given information and (or) objectives of the given problem.

(2） The problem representation method is only combined with CoT no matter zero-shot or few-shot. It would be better to explore the combination of the problem representation method with more distinct types of solution search methods like decomposition-based methods (e.g., L2M), to validate the generality the proposed problem representation method.

**Questions:**

Additional suggestions:

(1) In the “Abstract” and “Introduction” sections, it would be beneficial if the authors could emphasize some interesting experimental findings, such as PreCoT's enhanced robustness against irrelevant information and context sensitivity, as well as some types of errors are mitigated by PreCoT. Highlighting such results would help readers to quickly grasp that in which cases PreCoT is more effective and why it works.

(2) Regarding Figure 1, it would be better to visually distinguish the two stages more clearly. In the current version, "#1 Problem Representation Construction" and "#2 Solution Searching" on the left may be deem as simultaneous inputs of an LLM at the first sight. I recommend referring to Figure 1 in the Least-to-most paper (https://openreview.net/pdf?id=WZH7099tgfM) for inspiration on how to improve the visual separation.

(3) I noticed that in the paper, the implementation of PreCoT involves three separate prompts, resulting in three API calls. I'm curious if consolidating these three prompts into two or even one prompt could still achieve good performance. If so, such a consolidation could significantly reduce API calling costs and improve efficiency.

(4) In Figure 3, "Math World Problem" should be "Math Word Problems."

---

> ### Author Response · Authors · 2023-11-17
> **Response to Reviewer 9iGg (1/2)**
>
> We appreciate your insightful and constructive feedback. Also, thank you for highlighting the simplicity and effectiveness of our approach. In response to your comment, we've included further details and experimental data in the following sections. If you have any more feedback, suggestions, or questions, please don't hesitate to share them with us.
>
> ---
>
> **(1) Generality of our method.**
>
> We appreciate your constructive suggestion. We implement the Least-to-Most (LtM) prompting [1] as described in the original paper. Accordingly, we also implement Problem Representation Enhanced LtM (PreLtM) by attaching the problem representation construction stage to LtM. In PreLtM, LtM is grounded on a self-generated problem representation. We evaluated both baselines in the fair evaluation setup described in our paper using PaLM 2. We employed the most benchmarks used by the original LtM paper. Additionally, we added GSM-IC [2] to investigate the effectiveness of problem representation on robustness against irrelevant information you highlighted as our strength.
>
> |  | LtM | PreLtM |
> | --- | --- | --- |
> | GSM8K | 54.51 | 57.09 (+2.58) |
> | GSM-IC | 62.50 | 73.40 (+10.90) |
> | DROP (football) | 71.60 | 72.80 (+1.20) |
> | DROP (non-football) | 78.80 | 79.80 (+1.00) |
> | Last Letters | 50.40 | 48.53 (-1.87) |
>
> The results show that our approach can be incorporated with other solution searching methods, demonstrating the universal applicability and effectiveness. We observe consistent improvement on all arithmetic reasoning tasks, including additional DROP [3] benchmarks consistent with our main results.
>
> For the degradation in last letters concatenation (Last Letters), LLM often fails to concatenate the last letters of the given words in both LtM and PreLtM, while it can successfully extract the last letters of each word. Since our method aims to provide an effective interpretation of a problem rather than directly manipulating specific solution searching steps, the performance improvement may be insignificant for specific tasks where a particular step in the solution searching stage is the bottleneck (e.g., concatenation).
>
> Consistent with our main result, problem representation did not improve the performance on Last Letters in LtM, either. Please note that in zero-shot setting, where the solution structure of this task is not provided to LLMs, our PreCoT significantly boosts the performance on Last Letters (+15.06% in PaLM 2 and +10.00% in GPT-3).
>
> We will update the paper with this result after receiving further feedback.
>
> **(2) Previous similar works.**
>
> We carefully reviewed the papers you suggested and would like to provide what makes our paper different from the previous works.
>
> First, we’d like to provide a comprehensive summary of our approach. Our work enhance general solution searching performance of LLMs by grounding this process with a structural interpretation of a problem inspired by human problem-solving. This approach can be applied easily to various solution searching methods such as CoT and LtM.
>
> The papers you mentioned are a sort of solution searching method that involves the given information extraction. They used the extracted information to manipulate specific solving steps, such as filtering out irrelevant information [4] or verifying each reasoning step [5]. In particular, while the prompt used in [4] includes the instruction “understand the problem”, it might not be seen as a specific strategy to improve understanding of a problem. The core of our idea is not to simply extract specific elements of a question but to ground solution searching by structured interpretation of a problem observed in human problem-solving.
>
> **(3) Highlighting interesting experimental findings in the “Abstract” and “Introduction” sections.**
>
> We have incorporated your suggestions into our paper. We will further improve these sections to clarify our contribution after receiving more feedback across the board.
>
> **(4) Clarifying Figure 1.**
>
> We updated Figure 1 to reflect your suggestion for better visibility. We will follow up if you have further feedback on the presentation.
>
>
> **(5) Single-pass implementation of PreCoT.**
>
> Thanks for the constructive suggestion. We think that the PreCoT can be implemented in a single-pass. However, our main goal in this study is to investigate if the problem representation can enhance the solution process of LLMs. Therefore, we more focused on reliably generating problem representations without concerning the generation efficiency. That said, we will explore the consolidation you mentioned for the more efficient implementation.
>
> **(6) About the typo in the title of Figure 3.**
>
> Thank you for pointing it out. We corrected the typo.

---

> > ### Author Response · Authors · 2023-11-17
> > **Response to Reviewer 9iGg (2/2)**
> >
> > **References**
> >
> > [1] Zhou, D., Schärli, N., Hou, L., Wei, J., Scales, N., Wang, X., ... & Chi, E. H. (2022, September). Least-to-Most Prompting Enables Complex Reasoning in Large Language Models. In *The Eleventh International Conference on Learning Representations*.
> >
> > [2] Shi, F., Chen, X., Misra, K., Scales, N., Dohan, D., Chi, E.H., Scharli, N., & Zhou, D. (2023). Large Language Models Can Be Easily Distracted by Irrelevant Context. *International Conference on Machine Learning*.
> >
> > [3] Dua, D., Wang, Y., Dasigi, P., Stanovsky, G., Singh, S., & Gardner, M. (2019). DROP: A Reading Comprehension Benchmark Requiring Discrete Reasoning Over Paragraphs. *North American Chapter of the Association for Computational Linguistics*.
> >
> > [4] Wang, L., Xu, W., Lan, Y., Hu, Z., Lan, Y., Lee, R.K., & Lim, E. (2023). Plan-and-Solve Prompting: Improving Zero-Shot Chain-of-Thought Reasoning by Large Language Models. *Annual Meeting of the Association for Computational Linguistics*.
> >
> > [5] Ling, Z., Fang, Y., Li, X., Huang, Z., Lee, M., Memisevic, R., & Su, H. (2023). Deductive Verification of Chain-of-Thought Reasoning. *ArXiv, abs/2306.03872*.

---

> > > ### Author Response · Authors · 2023-11-22
> > >
> > > Dear Reviewer 9iGg,
> > >
> > > We sincerely appreciate your insightful and constructive feedback. We have carefully considered your comments and have tried to comprehensively address your questions in our response. As the discussion period approaches its conclusion, we kindly inquire if there are any additional questions or if our responses have sufficiently addressed your concerns. We would be happy to discuss any additional feedback you may have.

---

> > ### Comment · Reviewer_9iGg · 2023-11-22
> >
> > Dear Authors,
> >
> > Thank you for your efforts in addressing the reviews. However, I still find the technical novelty somewhat limited, especially in the context of ICLR. While the idea of using the psychological notion is promising, the implementation lacks the desired level of innovation. Based on my understanding, the implementation primarily involves extracting the given information and objectives of problems, while you argue that the idea is not to simply extract specific elements of a question.
> > I anticipate more comprehensive and in-depth research on simulating the "problem representation construction" in human problem-solving with an LLM. Drawing on the latest findings from relevant cognitive psychology research, as mentioned in the paper, may contribute to it.
> >
> > Additionally, the characterization of previous methods as "solution searching methods" might be questionable. These methods, at the very least, implicitly construct the question representation (i.e., able to figure out the given information and objective). Since without this implicit understanding, they would be unable to successfully solve any problems. The primary distinction may be that the previous methods belong to "implicit problem representation/understanding + explicit reasoning", whereas PreCOT involves "explicit problem representation/understanding + explicit reasoning".
> >
> > For the reasons above, I would like to keep my score for now.

---

> > > ### Author Response · Authors · 2023-11-23
> > > **Follow-up Response to Reviewer 9iGg**
> > >
> > > Thank you for the follow-up feedback.
> > >
> > > ---
> > >
> > > **(1) About the implementation.**
> > >
> > > We want to clarify that our approach is not merely about 'extracting specific information.' The essence of our idea lies in structuring the problem into two states and examining how this structured interpretation affects the reasoning performance of LLMs. We view the implementation as a means to validate our methodology, rather than the end result of the approach itself.
> > >
> > > Furthermore, we wish to highlight that problem representation involves more than just repeating information. Recent studies have indicated that even straightforward verbatim repetition can significantly enhance the reasoning abilities of LLMs [1]. In our supplemental experiment, we demonstrated that effective problem representation transcends mere repetition. For your reference, we put the result of the experiments below (R1: repetition once; R2: repetition twice).
> > >
> > > |  | R1 + CoT | PreCoT |
> > > | --- | --- | --- |
> > > | GSM8K | 57.62 | 61.79 (+4.17) |
> > > | GSM-IC | 65.35 | 75.50 (+10.15) |
> > > | SVAMP | 74.00 | 78.20 (+4.20) |
> > > | AQuA | 43.70 | 44.49 (+0.79) |
> > >
> > > |  | R2 + CoT | PreCoT |
> > > | --- | --- | --- |
> > > | GSM8K | 55.72 | 61.79 (+6.07) |
> > > | GSM-IC | 63.50 | 75.50 (+12.00) |
> > > | SVAMP | 74.00 | 78.20 (+4.20) |
> > > | AQuA | 38.58 | 44.49 (+5.91) |
> > >
> > > **(2) Without an implicit understanding on problem, LLMs would be unable to successfully solve any problems.**
> > >
> > > We would like to kindly highlight the following points:
> > >
> > > The phrase “Let’s think step by step” [2] constitutes an explicit instruction. Although reasoning problems typically necessitate an implicit, step-by-step thought process, it has been observed that LLMs demonstrate significantly improved reasoning when guided explicitly. Furthermore, complex questions, which implicitly entail answering a series of simpler, interconnected questions, are better addressed by LLMs using the Least-to-most prompting [3]. This approach notably enhances LLM performance on intricate reasoning tasks by explicitly directing them along this problem-solving trajectory. We believe these findings suggest that explicitly directing LLMs in problem-solving processes is an effective approach for enhancing their performance.
> > >
> > > ---
> > >
> > > **References**
> > >
> > > [1] Xu, X., Tao, C., Shen, T., Xu, C., Xu, H., Long, G., & Lou, J. G. (2023). Re-reading improves reasoning in language models. *arXiv preprint arXiv:2309.06275*.
> > >
> > > [2] Kojima, T., Gu, S. S., Reid, M., Matsuo, Y., & Iwasawa, Y. (2022). Large language models are zero-shot reasoners. *Advances in neural information processing systems*, *35*, 22199-22213.
> > >
> > > [3] Zhou, D., Schärli, N., Hou, L., Wei, J., Scales, N., Wang, X., ... & Chi, E. H. (2022, September). Least-to-Most Prompting Enables Complex Reasoning in Large Language Models. In *The Eleventh International Conference on Learning Representations*.

---

### Official Review · Reviewer_c8na · 2023-11-02

**Soundness:** 2 fair
**Presentation:** 3 good
**Contribution:** 2 fair
**Rating:** 3
**Confidence:** 4

**Summary:**

This work introduces a new prompting method aiming at improving and extending Chain of Thought (CoT) prompting.
Drawing inspiration from cognitive psychology, it proposes to structure prompting according to human problem representation, specifying a problem as the initial state and the objective of the problem as the goal state.
Evaluation has been performed for two major LLM (PaLM2 and GPT-3) on a number of reasoning benchmarks, ranging from arithmetic (GSM8K) to symbolic and common sense reasoning. During evaluation, PRECoT was compared to CoT under both Zero-shot and Few-shot prompting. Results suggest that PRECoT improves performance in the majority of tested conditions, while in-depth analysis suggests one possible mechanism for improved performance via the reduction in major semantic-logical errors.

**Strengths:**

The paper has clear objectives and is generally well-structured, with a good balance between main article and supplementary material which is useful to understand the details of the prompting method.
Despite the amount of recent work on prompting variants, it can still be considered original work in that the method is not immediately reducible to one of the pre-existing CoT or CoT-like approaches.
The strongest element of the paper is the amount of experimental work and the clarity of presentation of its results, especially Tables 1, 3, 5.

**Weaknesses:**

The paper repeatedly mentions human problem-solving as a rationale and an inspiration for the approach, which raises two independent issues. The first one is the lack of convincing and up to date backing for the rather central claim that humans actually decompose problems as suggested in the paper. Despite referring to "accumulated insights", references in section 2 are by no means recent ones, nor would some of them be considered to pertain to cognitive psychology. Secondly, whether seeking inspiration from human problem solving would actually improve LLM reasoning remains a debated issue and, to say the least, the jury is still out on this topic. Some similarities between human an LLM reasoning, such as step by step decomposition, tend to be rather simplistic, while more rigorous findings, such as similarity in the influence of content over reasoning (as per the Wason selection task [Dasgupta et al., 2022]) may not scale up to all forms of LLM reasoning.
The other issue is simply whether the PRECoT method actually works by the hypothesized mechanism of problem decomposition, rather than through a clarification of questions that would facilitate content-based inference through a better specification of relationships between 'variables'.
The moderate effect on StrategyQA and CSQA is interpreted as a difficulty in constructing good representations, but this could be seen as self-referential, and alternative explanations could be suggested based on the above.

Dasgupta, I., Lampinen, A.K., Chan, S.C., Creswell, A., Kumaran, D., McClelland, J.L. and Hill, F., 2022. Language models show human-like content effects on reasoning. arXiv preprint arXiv:2207.07051.

**Questions:**

How is performance affected when not including "Let’s think step by step." in the PRECoT case?
How would you normalize the experiment for the additional length of PRECoT prompts (whether it implies additional information or information repetition in different places)?
Why did you not use GPT-4?

---

> ### Author Response · Authors · 2023-11-17
> **Response to Reviewer c8na (1/2)**
>
> Thank you for your thorough and insightful review. We also appreciate your comments on the clarity and originality of our paper. We have carefully considered your comments and have included further explanations and additional experimental data in the following sections to address your concerns. We are keen to hear any further feedback, suggestions, or questions you might have.
>
> ---
>
> **(1) The references in Section 2 are not recent ones.**
>
> We believe that a fundamental observation lasts for a long period of time. For instance, recent academic publications cite the references in Section 2 as foundational to their work. In particular, Gick’s model [1], which we mainly referenced, is cited in several studies published in 2023. [2] mentioned this study as a foundational study in human problem-solving. Also, the ideas of references were derived from cognitive psychology in a broad sense, focusing on the problem-solving application.
>
> **(2) Whether taking inspiration from human problem-solving can improve LLM reasoning remains a debated issue.**
>
> While many works have been exploring the potential of LLMs on reasoning tasks, borrowing ideas from human behavior studies [3, 4], this remains an open question. In line with this direction, we investigated the effectiveness of problem representation on solution searching in LLMs. Please note that our research does not aim to make LLM reasoning identical to human reasoning. Instead, we focus on adapting observations from human problem-solving that could benefit LLMs to enhance their performance in general reasoning tasks.
>
> **(3) The effectiveness of problem representation may not scale up to all forms of LLM reasoning.**
>
> We acknowledge the importance of the scalability of our approach. We also recognize that problem representation might not benefit all forms of LLM reasoning. However, we respectfully point out that due to the vast and diverse nature of reasoning abilities, it is nearly impossible to test our method across every potential reasoning task in a single study. Despite this limitation, we endeavored to conduct as comprehensive an evaluation as possible within practical constraints.
>
> We investigated the effectiveness of problem representation in LLMs, focusing on arithmetic, commonsense, and symbolic reasoning tasks. These categories encapsulate the core reasoning abilities typically expected from an LLM and align with the scopes of many existing studies in this domain [3, 5, 6, 7, 8]. To further validate our approach rigorously, we expanded the scale of our evaluation to include 15 benchmarks, which is beyond the usual scope.
>
> Evaluating different techniques in this field, including ours, against benchmarks designed to explore intrinsic similarities between human and LLM reasoning could be an intriguing direction for future research. However, the aim of our study is not to investigate these similarities. Instead, we focus on assessing our approach's utility in general reasoning tasks for LLMs.
>
> **(4) Does the problem representation imply additional information, or is the information repeated in a different place?**
>
> We appreciate your thoughtful question. In response to your question, we compare PreCoT with variants replacing problem representation with question repetition (information repetition). The results show that problem representation is significant beyond the effect of simple information repetitions (R1: repetition once; R2: repetition twice).
>
> |  | R1 + CoT | PreCoT |
> | --- | --- | --- |
> | GSM8K | 57.62 | 61.79 (+4.17) |
> | GSM-IC | 65.35 | 75.50 (+10.15) |
> | SVAMP | 74.00 | 78.20 (+4.20) |
> | AQuA | 43.70 | 44.49 (+0.79) |
>
>
> |  | R2 + CoT | PreCoT |
> | --- | --- | --- |
> | GSM8K | 55.72 | 61.79 (+6.07) |
> | GSM-IC | 63.50 | 75.50 (+12.00) |
> | SVAMP | 74.00 | 78.20 (+4.20) |
> | AQuA | 38.58 | 44.49 (+5.91) |

---

> ### Author Response · Authors · 2023-11-17
> **Response to Reviewer c8na (2/2)**
>
> **(5) How is performance affected when not including "Let’s think step by step." in the PRECoT case?**
>
> We appreciate your thoughtful question. We initially put “Let’s think step by step.” in both CoT and PreCoT because the previous works showed that adding such instruction significantly boosts the performance of CoT [5, 6]. In response to your question, we test both CoT and PreCoT without “Let’s think step by step.”. The results show that PreCoT outperforms CoT regardless of the presence or absence of this instruction. We will update the appendix section with these results.
>
> |  | CoT (no step-by-step) | PreCoT (no step-by-step) |
> | --- | --- | --- |
> | GSM8K | 41.62 | 55.19 (+13.52) |
> | GSM-IC | 36.30 | 70.05 (+33.75) |
> | SVAMP | 74.90 | 76.40 (+1.50) |
> | AQuA | 38.98 | 42.52 (+3.54) |
>
>
> **(6) PreCoT may facilitate content-based inference by better specifying the relationships between variables, rather than by working through the problem decomposition.**
>
> The core of our idea is not to extract specific elements of a question but to ground solution searching by structured interpretation of a problem observed in human problem-solving. Actually, “better specification of relationships between variables” is a desirable ability for problem solvers. So we believe that is not the means, but the end.
>
> If we've misunderstood something in this part, we'd appreciate clarification.
>
> **(7) Why did you not use GPT-4?**
>
> We used the state-of-the-art models available to us, considering the scale of our evaluation. PaLM 2 is the most recent LLM proposed by Google, and text-davinci-003 is also one of the most recent models among the GPT-3 splits. We employ the major LLMs from different groups (with varying architectures, training corpora, and training objectives) to rigorously validate the general effectiveness of our method. These models are actively used in this domain. GPT-4 could not be used for our evaluation as it would entail prohibitive costs (both in terms of time and money) for our extensive evaluation. Instead, we tried rigorously validating our method with various benchmarks using the latest, cost-effective models.
>
> ---
>
> **References**
>
> [1] Gick, M. L. (1986). Problem-solving strategies. *Educational psychologist*, *21*(1-2), 99-120.
>
> [2] Reed, S. K. (2016). The structure of ill-structured (and well-structured) problems revisited. *Educational Psychology Review*, *28*, 691-716.
>
> [3] Zhou, D., Schärli, N., Hou, L., Wei, J., Scales, N., Wang, X., ... & Chi, E. H. (2022, September). Least-to-Most Prompting Enables Complex Reasoning in Large Language Models. In *The Eleventh International Conference on Learning Representations*.
>
> [4] Webb, T., Holyoak, K. J., & Lu, H. (2023). Emergent analogical reasoning in large language models. *Nature Human Behaviour*, *7*(9), 1526-1541.
>
> [5] Kojima, T., Gu, S. S., Reid, M., Matsuo, Y., & Iwasawa, Y. (2022). Large language models are zero-shot reasoners. *Advances in neural information processing systems*, *35*, 22199-22213.
>
> [6] Fu, Y., Peng, H., Sabharwal, A., Clark, P., & Khot, T. (2022, September). Complexity-Based Prompting for Multi-step Reasoning. In *The Eleventh International Conference on Learning Representations*.
>
> [7] Wei, J., Wang, X., Schuurmans, D., Bosma, M., Xia, F., Chi, E. H., ... & Zhou, D. (2022, May). Chain-of-Thought Prompting Elicits Reasoning in Large Language Models. In *Advances in Neural Information Processing Systems*.
>
> [8] Ling, Z., Fang, Y., Li, X., Huang, Z., Lee, M., Memisevic, R., & Su, H. (2023). Deductive Verification of Chain-of-Thought Reasoning. *ArXiv, abs/2306.03872*.

---

> > ### Author Response · Authors · 2023-11-22
> >
> > Dear Reviewer c8na,
> >
> > We sincerely appreciate your insightful and constructive feedback. We have carefully considered your comments and have tried to comprehensively address your questions in our response. As the discussion period approaches its conclusion, we kindly inquire if there are any additional questions or if our responses have sufficiently addressed your concerns. We would be happy to discuss any additional feedback you may have.

---

> > > ### Comment · Reviewer_c8na · 2023-11-22
> > > **Acknowledgement of Authors Response**
> > >
> > > Dear Authors,
> > > Thank you for having provided additional results in response to my questions. The results without the "step-by-step" component are interesting. I'm slightly less convinced by the use of repetition, assuming it is verbatim repetition, as it might not enhance the prompting strategy in a way that facilitates, e.g. analogical reasoning. In any case, I am not asking for further experiments or results on this specific topic at this late stage.
> > > There is a slight contradiction in your response to question (2), as seeking inspiration from human reasoning seemed to feature in the rationale section of the paper.
> > > Overall, your response only marginally changes my appraisal of the paper.

---

> > > > ### Author Response · Authors · 2023-11-23
> > > > **Follow-up Response to Reviewer c8na**
> > > >
> > > > We appreciate the follow-up feedback!
> > > >
> > > > ---
> > > >
> > > > **(1) Repetition might not enhance the prompting strategy in a way that facilitates.**
> > > >
> > > > Recent literature demonstrates that even simple verbatim repetition can enhance the reasoning performance of LLMs [1]. Therefore, in our additional experiment, we aimed to demonstrate that problem representation involves more than just repeating information; it's about interpreting the problem, including the given information and the objective.
> > > >
> > > > **(2) Seeking inspiration from humans.**
> > > >
> > > > We would like to kindly point out that seeking inspiration from humans does not equate to perceiving LLMs as human. We believe that there is nothing wrong with gaining insights by observing human behavior, especially since it falls within the open domain.
> > > >
> > > > ---
> > > > **References**
> > > >
> > > > [1] Xu, X., Tao, C., Shen, T., Xu, C., Xu, H., Long, G., & Lou, J. G. (2023). Re-reading improves reasoning in language models. *arXiv preprint arXiv:2309.06275*.

---

### Official Review · Reviewer_Hado · 2023-11-06

**Soundness:** 2 fair
**Presentation:** 3 good
**Contribution:** 2 fair
**Rating:** 3
**Confidence:** 4

**Summary:**

This paper presents PRECoT, a prompting strategy that prompts the LLM on extracting the problem information before coming up with the solution. The results show that on multiple benchmarks PRECoT is a better prompting strategy than CoT.

**Strengths:**

- The paper is well-written and easy to follow.
- The authors perform experiments on multiple benchmarks and utilize the state-of-the-art models.
- Error analysis experiments are performed to show the importance of problem representation.

**Weaknesses:**

The claim that PreCOT helps LLM enhance its reasoning doesn’t make sense to me as the LLMs are more or less performing approximate retrieval based on the training data that has been fed into them. When these datasets include not only solutions but also the derivative paths leading to them, methodologies like PreCOT might improve an LLM's ability to generate more probable outputs, thus appearing to solve problems more effectively [1]. However, given the vast and varied nature of the content available on the web, we have no clear idea as to what the entire web contains.  In my view, to substantiate the claim that LLMs possess the ability to reason, some kind of diagonalization should be done in the kind of benchmarks that the approach is being tested on. An example of such diagonalization is the obfuscation method that the authors in [2] propose, while testing the planning abilities of LLMs. My primary concern is that the benchmarks used in the proposed work are susceptible to be part of or mirror the training data of the LLM. While testing these benchmarks are useful, the claims made in the paper can be upheld if there is some form of diagonalization in the benchmarks that are being evaluated.

[1] McCoy, R. T., Yao, S., Friedman, D., Hardy, M., & Griffiths, T. L. (2023). Embers of autoregression: Understanding large language models through the problem they are trained to solve. arXiv preprint arXiv:2309.13638.

[2] Valmeekam, K., Marquez, M., Sreedharan, S., & Kambhampati, S. (2023). On the Planning Abilities of Large Language Models--A Critical Investigation. arXiv preprint arXiv:2305.15771.

**Questions:**

1. How many examples were given in the few-shot setting?

2. (Minor) Why do you call it problem representation? Isn’t it problem information that is being extracted? I feel like calling it problem representation might make the reader believe that LLM is extracting how the problem is being represented rather than what are the crucial information pieces in the question.

---

> ### Author Response · Authors · 2023-11-17
> **Response to Reviewer Hado (1/2)**
>
> We appreciate your thoughtful and detailed feedback. Also, thank you for highlighting our paper's clarity and experimental setup. We carefully reviewed your comments and added more details and data in the following sections. Please feel free to share any additional feedback, suggestions, or questions.
>
> ---
>
> **(1) Concerns about reasoning ability of LLMs.**
>
> We acknowledge some concerns and different perspectives in the community on whether LLMs have reasoning ability. While many researchers have been exploring the topic to answer the controversial question, it remains open [1]. Our primary goal in this study is not to conclude the debate by firmly insisting that LLMs possess reasoning ability. Instead, we aim to further explore the potential of LLMs in reasoning tasks by applying ideas borrowed from well-established literature on human behavior.
>
> **(2) LLMs are more or less performing approximate retrieval.**
>
> Even if we call it retrieval, LLMs should go through some steps to find the desirable information, not just directly spotting it. It involves logically combining necessary information. We used various benchmarks that are designed to evaluate the reasoning ability of LLMs to validate our approach rigorously. Those benchmarks were extensively used in prior studies [2, 3, 4, 5] to assess the specific reasoning abilities of LLMs.
>
> In particular, the mathematical and symbolic reasoning problems we covered in our study cannot be regarded as mere information retrieval tasks. Specifically, GSM-IC [6] incorporates irrelevant information to distract LLMs, and SVAMP [7] requires LLMs to consider the context of a problem rather than relying on simple heuristics.
>
> For example, the question in Figure 2 of our paper inquires about the total gain derived from reselling some magazines. In this case, LLMs must distinguish the relevant information from the question and logically combine these to reach the correct answer.
>
> While we respect and acknowledge your concerns about the reasoning of LLMs, we would like to emphasize that the benchmarks we used are devised to evaluate the reasoning ability [6, 7, 8, 9, 10]. Our method shows competitive performance on them.
>
> **(3) Primary concern: Benchmarks used in the proposed work are susceptible to be part of or mirror the training data of the LLM.**
>
> The data used to train the LLMs employed in our experiments includes documents only up until mid-2021. This means most of the benchmarks we used, such as GSM8K (released in October 2021) [10], GSM-IC (released in January 2023) [6], and all tasks from Big Bench (released in June 2022) [8, 9], were not part of the training data for these LLMs due to their release dates being subsequent to their data cutoff. Furthermore, the Big Bench tasks explicitly prohibit their inclusion in the pre-training corpora of LLMs.
>
> **(4) How many examples were given in the few-shot setting?**
>
> To ensure fairness, we reused the examples from previous works [5, 9] in most cases. The number of examples comprising each prompt is as follows.
>
> | Benchmarks | # of examples |
> | --- | --- |
> | SocialIQA, All Big Bench tasks | 3 |
> | Last Letter Concatenation | 4 |
> | StrategyQA, Coin Flips | 6 |
> | CommonsenseQA | 7 |
> | All arithmetic reasoning benchmarks | 8 |
>
>
> **(5) Why do you call it problem representation?**
>
> The essence of our concept is not merely to extract specific components from a question. Instead, our approach is about grounding solution searching process of LLMs with problem representation, a structured interpretation of a problem comprising the initial (given information) and goal states (goal information) [11]. We will make it clearer in the revised version.

---

> > ### Author Response · Authors · 2023-11-17
> > **Response to Reviewer Hado (2/2)**
> >
> > **References**
> >
> > [1] Huang, J., & Chang, K.C. (2023). Towards Reasoning in Large Language Models: A Survey. In *Findings of the Association for Computational Linguistics: ACL 2023* (pp. 1049–1065). Association for Computational Linguistics.
> >
> > [2] Zhou, D., Schärli, N., Hou, L., Wei, J., Scales, N., Wang, X., ... & Chi, E. H. (2022, September). Least-to-Most Prompting Enables Complex Reasoning in Large Language Models. In *The Eleventh International Conference on Learning Representations*.
> >
> > [3] Kojima, T., Gu, S. S., Reid, M., Matsuo, Y., & Iwasawa, Y. (2022). Large language models are zero-shot reasoners. *Advances in neural information processing systems*, *35*, 22199-22213.
> >
> > [4] Ling, Z., Fang, Y., Li, X., Huang, Z., Lee, M., Memisevic, R., & Su, H. (2023). Deductive Verification of Chain-of-Thought Reasoning. *ArXiv, abs/2306.03872*.
> >
> > [5] Wei, J., Wang, X., Schuurmans, D., Bosma, M., Xia, F., Chi, E. H., ... & Zhou, D. (2022, May). Chain-of-Thought Prompting Elicits Reasoning in Large Language Models. In *Advances in Neural Information Processing Systems*.
> >
> > [6] Shi, F., Chen, X., Misra, K., Scales, N., Dohan, D., Chi, E.H., Scharli, N., & Zhou, D. (2023). Large Language Models Can Be Easily Distracted by Irrelevant Context. *International Conference on Machine Learning*.
> >
> > [7] Patel, A., Bhattamishra, S., & Goyal, N. (2021, June). Are NLP Models really able to Solve Simple Math Word Problems?. In *Proceedings of the 2021 Conference of the North American Chapter of the Association for Computational Linguistics: Human Language Technologies* (pp. 2080-2094).
> >
> > [8] Srivastava, A., Rastogi, A., Rao, A., Shoeb, A. A. M., Abid, A., Fisch, A., ... & Wang, G. X. (2023). Beyond the Imitation Game: Quantifying and extrapolating the capabilities of language models. *Transactions on Machine Learning Research*.
> >
> > [9] Suzgun, M., Scales, N., Schärli, N., Gehrmann, S., Tay, Y., Chung, H., Chowdhery, A., Le, Q., Chi, E., Zhou, D., & Wei, J. (2023). Challenging BIG-Bench Tasks and Whether Chain-of-Thought Can Solve Them. In *Findings of the Association for Computational Linguistics: ACL 2023* (pp. 13003–13051). Association for Computational Linguistics.
> >
> > [10] Cobbe, K., Kosaraju, V., Bavarian, M., Chen, M., Jun, H., Kaiser, L., Plappert, M., Tworek, J., Hilton, J., Nakano, R., Hesse, C., & Schulman, J. (2021). Training Verifiers to Solve Math Word Problems. *ArXiv, abs/2110.14168*.
> >
> > [11] Gick, M. L. (1986). Problem-solving strategies. *Educational psychologist*, *21*(1-2), 99-120.

---

> > > ### Author Response · Authors · 2023-11-22
> > >
> > > Dear Reviewer Hado,
> > >
> > > We sincerely appreciate your insightful and constructive feedback. We have carefully considered your comments and have tried to comprehensively address your questions in our response. As the discussion period approaches its conclusion, we kindly inquire if there are any additional questions or if our responses have sufficiently addressed your concerns. We would be happy to discuss any additional feedback you may have.

---

### Meta-Review · Area_Chair_uN2z · 2023-12-14

**Metareview:**

Inspired by CoT, the paper proposes PreCoT a multi-step prompting strategy for solving complex reasoning tasks. First, PreCoT prompts the LLM to break down the input question into (1) Given Information; (2) Objective; and then (3) ask the model to solve the question.
PreCoT evaluated on PaLM2 and GPT-3 and across a wide set of multi-step reasoning tasks including arithmetic, commonsense, and symbolic reasoning.

The main result is that PreCoT tends to outperform CoT on many tasks. That is, the result is mixed (Table 1, 3, 5 contains a few settings where PreCoT underperforms the baseline CoT).
Some reviewers (`9iGg`) were concerned with the novelty of the approach and some reviewers (`9iGg` and `c8na `) requested a more fundamental take-away that motivates the method stronger. I also agree with reviewer (`2rHa`) that a demonstration on GPT-4 should also strengthen the paper and makes the impact of the method clearer.

For a more complete comparison to the SotA, the authors might want to try comparing with Self-Consistency CoT and Tree of Thoughts (more recent methods that outperform CoT). Also, multiple approaches for Step 1 (i.e. breaking down the input question into Information Given + Objectives) can be explored to understand the impact of this style of prompting.

I think all the above are valid concerns from the reviewers since it has been 1.5 years since CoT (Let's think step by step) was introduced and there have been many LLM prompting approaches invented. Given the current state of the manuscript, the impact of PreCoT is a bit weak. Therefore, I recommend `reject`.

The AC thinks that this work in its current form may fit more to the new CoLM conference (https://colmweb.org/) than ICLR.

**Justification For Why Not Higher Score:**

The method lacks a more fundamental motivation. The results are slightly mixed and the latest LLMs like GPT-4 are not tested.N

**Justification For Why Not Lower Score:**

N/A

---

### Decision · Program_Chairs · 2024-01-16

Reject